# Activated Carbon from Yam Peels Modified with Fe₃O₄ for Removal of 2,4-Dichlorophenoxyacetic Acid in Aqueous Solution

**Udualdo Herrera-García** [1] , **Jefferson Castillo** [1], **David Patiño-Ruiz** [2] , **Ricardo Solano** [2] and **Adriana Herrera** [1,2,*]

1 Chemical Engineering Program, Nanomaterials and Computer Aided Process Engineering Research Group, University of Cartagena, Cartagena 130015, Colombia; uherrerag@unicartagena.edu.co (U.H.-G.); jcastillom4@unicartagena.edu.co (J.C.)
2 Doctorate in Engineering Program, Nanomaterials and Computer Aided Process Engineering Research Group, University of Cartagena, Cartagena 130015, Colombia; dpatinor@unicartagena.edu.co (D.P.-R.); rsolanop@unicartagena.edu.co (R.S.)
\* Correspondence: aherrerab2@unicartagena.edu.co

**Abstract:** The removal of organic pollutants from water sources can be enhanced using suitable adsorbents. The aim of this research was to study the adsorption capacity and potential reuse of a magnetic adsorbent prepared from agricultural wastes of yam peels (*Dioscorea rotundata*) for 2,4-dichlorophenoxyacetic (2,4-D) acid removal. The procedure was performed through carbonization and activation at 400 and 500 °C, respectively. Then, the as-prepared activated carbon (AC) was chemically modified using magnetite (Fe₃O₄) nanoparticles. The AC and magnetic activated carbon (MAC) were characterized and then used for batch adsorption and regeneration tests at different pH, initial concentrations of 2,4-D, and temperature. AC and MAC were showed to have microporous structures with surface areas of 715 and 325 m²/g, respectively. Superparamagnetic behavior was observed for MAC with a saturation magnetization of 6 emu/g. The results from the batch experiments showed higher adsorption capacity at high initial concentration of 2,4-D, low pH, and room temperature. The thermodynamic parameters indicated that the experiments proceeded as exothermic and spontaneous adsorption. Our findings also showed that MAC can be separated from the water medium through a facile magnetic procedure, and from regeneration experiments, MAC showed better results with 60% of its initial adsorption capacity after five cycles. Hence, MAC was found to be a promising alternative adsorbent of pesticides in water.

**Keywords:** yam peel; carbonization; activation; magnetic activated carbon; 2,4-dichlorophenoxyacetic acid

## 1. Introduction

The rapidly growing world population has become an important matter regarding agricultural activities. In the last century, the production of crops has doubled due to the need for meeting the demand for food. Although technologies have arisen to improve the performance of lands for harvesting, the wide use of pesticides such as herbicides is still the most effective and with low cost. Herbicides have been catalogued as one of the most hazardous pollutants to the environment due to their toxic effects after prolonged exposure. Residues of this pollutant have been found in food and drinking water, which represents a risk to human health leading to eye and skin irritation, headaches, dizziness, and nausea, and even to chronic illnesses such as cancer, asthma, and diabetes [1,2]. After spraying, a large amount of herbicides ends up in water sources such as rivers which, through rainfall, can migrate

easily and contaminate other sources such as underground water [3,4]. 2,4-dichlorophenoxyacetic acid (2,4-D) is a common herbicide with high efficiency and low cost which has been widely used in agribusiness for the neutralization of weeds in different crops [5]. 2,4-D is labeled by different international agencies as a non-biodegradable carcinogenic and mutagenic compound [6], and it has been found in the water sources of different countries [7] such as Argentina (1.9 mg/L) [8], Colombia (0.001 mg/L) [9], Canada (0.002775 mg/L), Iran (0.016 mg/L), and Ireland ($1 \times 10^{-6}$ mg/L) [10]. 2,4-D is among the 10 most used active agents in pesticides in the United States, and it is often used for essential crops such as sugarcane, corn, wheat, oats, and barley [11].

Several physical and chemical methods have been adopted to reduce the environmental impact of herbicides, including catalytic degradation, biological oxidation, aerobic degradation, adsorption, nanofiltration, liquid extraction, and solid-phase extraction as the most common for 2,4-D removal [5,12,13]. Adsorption has been widely applied due to its efficiency, capacity, and applicability on a large scale [14]. Activated carbon is an efficient adsorbent for the removal of organic pollutants in aqueous solutions due to its physicochemical properties, such as high surface area and microporosity [5]. Additionally, agroindustrial wastes are being used for the production of activated carbons, allowing us to solve two environmental problems in one approach through the transformation of these natural sources into adsorbent materials [15–17]. Adsorption of 2,4-D using activated carbon prepared from natural sources has been reported to show good adsorption capacity [5,6,18]. Among these reports, adsorption capacities were found using activated carbon from date stones (238.10 mg/g) [19], potassium carbonate (22.84 mg/g) [20], langsat empty fruit (261.2 mg/g) [18], and groundnut shells (250 mg/g) [6].

One of the major disadvantages of these adsorbents is the recovery process from an aqueous medium. Hence, the modification of activated carbon with magnetic nanoparticles has recently attracted great attention from the scientific community. This type of modification aims to facilitate the recovery of the material after the adsorption of pesticides in aqueous media. Besides this, the magnetic separation process has been shown to be simple and efficient and with low cost, taking into account that these nanoparticles present low toxicity in an aqueous medium [21–23].

White yam (*Dioscorea rotundata*) is a medium-grown crop tuber grown in the northern part of Colombia, where it is widely used in the common diet of the inhabitants of this area of the country and is also an export product to the U.S. and European markets that generates for the country revenues of more than U.S. $2.5 M annually [24,25]. However, the industrial production and consumption of white yam generates a large amount of waste that is improperly disposed of in the environment. Since the white yam is a starch-rich tuber and the peel is composed of more than 45% elemental carbon [24], the preparation of activated carbon as an adsorbent material with good physicochemical properties is possible.

This research aimed to study the adsorption potential of a magnetic adsorbent and its reuse capability. We hereby report the preparation of magnetic activated carbon (MAC) using white yam peels as a natural source and magnetite nanoparticles, with the aim to obtain an adsorbent that can be easily recovered and reused for the removal of 2,4-D in aqueous solution. We found that the morphological and physicochemical properties of the MAC were similar to those of carbons synthesized using other natural sources. Additionally, parameters such pH, initial concentration, and temperature were evaluated in terms of the adsorptive capacity and regeneration percentage of the MAC. From the adsorption experiments, the results showed that MAC is a promising adsorbent for the removal of organic pollutants in the environmental remediation field.

## 2. Materials and Methods

### 2.1. Materials

*Dioscorea rotundata (D. rotundata)* peels were collected from a local company in Cartagena, Colombia. Hydrochloric acid 38% (HCl), sulfuric acid 36% ($H_2SO_4$), ethanol 95%, nitric acid 65% ($HNO_3$), and sodium hydroxide (NaOH) were purchased from Chemí. Iron(II) chloride tetrahydrate

95% ($FeCl_2 \cdot 4H_2O$) was acquired from Alfa Asear® (Ward Hill, MA, USA). Potassium hydroxide 85% (KOH), orthophosphoric acid 85% ($H_3PO_4$), and iron(III) chloride hexahydrate 95% ($FeCl_3 \cdot 6H_2O$) were purchased from PanReac® (Castellar del Vallès, Barcelona, Spain). A commercial formulation of the herbicide 2,4-dichlorophenoxyacetic (2,4-D) acid was acquired from Invesa® (Envigado, Antioquia, Colombia). Distilled water was used in all experiments.

### 2.2. Preparation of Activated Carbon (AC) and Magnetic Activated Carbon (MAC)

#### 2.2.1. Synthesis of Magnetite ($Fe_3O_4$) Nanoparticles

$Fe_3O_4$ nanoparticles were synthesized by a chemical co-precipitation method [26,27]. In a typical experiment, 4.86 g $FeCl_3 \cdot 6H_2O$ and 1.79 g $FeCl_2 \cdot 4H_2O$ were dissolved in 50 mL of distilled water, separately. Then, the iron salt solutions were mixed and then heated at 70 °C under 120 rpm mechanical stirring. Afterwards, 6 M NaOH solution was added dropwise at 10 mL/min to adjust to pH 10. The reaction was carried out for 1 h, and the pH was maintained by dropwise adding 30 mL of distilled water and 2 M NaOH solution at 5 mL/min each 10 min. The black precipitate was separated magnetically and washed two times with 200 mL of distilled water each and once with 200 mL of ethanol for 5 min. The $Fe_3O_4$ nanoparticles were dried in an oven at 70 °C overnight.

#### 2.2.2. Preparation of Activated Carbon (AC) from *D. Rotundata* Peels

Accordingly, *D. Rotundata* peels were separated from the pulp and washed using 0.01 M HCl solution and 200 mL of distilled water and then dried in an oven at 110 °C for 7 h. The biomass was ground and sieved using a no. 200 mesh. The particulate matter was added in a 10% w/v solution of 0.5 M $H_2SO_4$. Afterwards, the particulate matter was washed three times with 200 mL of distilled water each to neutralize the pH and then dried in an oven at 110 °C for 7 h.

Recently, the effects of residence time, temperature, and oxygen concentration on the carbonization of biomass have been studied; the reports show that the best conditions are attributed to a short residence time, temperatures below 420 °C, and low oxygen concentrations. Therefore, the oxidation of biomass is mainly due to exposure to natural atmosphere for a prolonged residence time [28]. Although the thermal treatment of biomass is carried out at low temperatures, the residence time is usually extended for long periods above 90 min, increasing the exposure to the natural atmosphere. Additionally, some reports indicated that variation in the total mass loss of the material during carbonization under natural atmosphere was similar to that using inert atmospheres [29]. We report the preparation of carbons through carbonization at 400 °C for a short residence time (heating 5 °C/min until 400 °C), which may contribute to low oxidation of the biomass. Accordingly, the particulate matter was treated with $H_3PO_4$ solution considering a 1:1 mass ratio at 80 °C for 6 h under 180 rpm mechanical stirring. The material treated was separated from the $H_3PO_4$ solution and dried in an oven at 110 °C for 4 h and subsequently carbonized in a muffle furnace at 400 °C under natural atmosphere, with a heating rate of 5 °C/min. Once the temperature was reached, the sample was taken out of the muffle furnace and cooled down at room temperature for 1 h in order to decrease the oxidation of carbon [28,29]. The as-prepared carbon was washed with 200 mL of distilled water, put in contact with 200 mL of ethanol for 24 h, and dried in an oven at 110 °C for 7 h. Carbon was activated using a KOH solution considering a 1:2 mass ratio, under constant stirring for 1 h, and then placed in a muffle furnace at 500 °C with a rate of 5 °C/min. Finally, the as-prepared AC was washed with 200 mL of distilled water and dried in an oven at 110 °C for 24 h.

#### 2.2.3. Preparation of MAC Using Magnetite $Fe_3O_4$ Nanoparticles

Considering a 2:3 mass ratio, 4 g of AC was initially added into a flask placed in an ultrasound bath at 80 °C which contained an $HNO_3$ solution and left to react for 3 h. The treated AC was then separated from the solution and dried at room temperature. Afterwards, the AC treated was added into a solution with $Fe_3O_4$ nanoparticles using a 2:3 mass ratio and was then placed in an ultrasound

bath at 80 °C for 1 h [21]. The as-prepared MAC was filtered and washed three times with 200 mL of distillated water each and once with 200 mL of ethanol. Finally, the MAC was dried in an oven at 80 °C for 24 h.

## 2.3. Characterization of Materials

The AC and MAC were characterized to determine their physicochemical and morphological properties. Scanning electron microscopy and energy-dispersive X-ray spectroscopy (SEM-EDS) techniques were performed to obtain images and the elemental composition using a LEO-LEICA Stereoscan 440 (Leica, Mannheim, Germany) coupled to a Bruker AXS XFlash Detector 4010 (Bruker, Berlin, Germany) with an acceleration voltage between 15 and 20 KV. Functional groups were identified using an IRAffinity-1 FTIR SHIMADZU (SHIMADZU, Kyoto, Japan). X-ray diffraction (XRD) patterns were acquired using a PANalytical model X´pert Instrument (Malvern, Almelo, Netherlands), with Cu Kα radiation (λ = 1.54 A°) at 450 KV and 400 mA; the intensities were measured at diffraction 2θ from 0° to 120° for carbons and 0° to 90° for magnetite nanoparticles. The surface area was evaluated by nitrogen adsorption at −196 °C in a Micromeritic Model ASAP 2020 Plus. A vibrating sample magnetometer (VSM) (Micromeritics Instrument Corp., Norcross, GA, USA) was used to obtain the magnetization curves at 300 K in an applied field of 50 KOe. The point of zero charge (pH$_{pzc}$) was determined experimentally using distilled water at different initial pH values (2, 4, 6, 8, 10, and 12), where the samples were put in contact for 48 h under 180 rpm mechanical stirring. The initial and final pH values of the solution were measured and plotted together with a straight line at 45°; the intercept of the curve with the straight line was called pH$_{pzc}$ for each carbon [22,30].

## 2.4. Batch Adsorption Study

Adsorption experiments for 2,4-D removal using AC and MAC were performed via a batch system in which the effects of initial concentration, contact time, pH, and temperature were studied. A stock solution of 2,4-D was previously prepared at a concentration of 300 ppm. Then, dilutions at 50, 100, 150, and 200 ppm from the stock solution were prepared in distilled water. For all the adsorption experiments, 0.1 g quantities of AC and MAC were added into 50 mL of 2,4-D solution, placed on a shaker at 120 rpm, and then stored in amber vials for further analysis using UV–vis and HPLC techniques. The concentration of 2,4-D solution before and after batch experiments was analyzed in a single-beam UV–vis spectrophotometer (BK-UV1900, Qingdao, China) at a wavelength of 284 nm [5]. In order to corroborate the results, the kinetic curve at 25 °C for AC was further analyzed using HPLC (LC 1260 Infinity II, Agilent Technologies, Santa Clara, CA, USA) coupled with UV (1260 Infinity II Variable Wavelength Detector, Agilent, Santa Clara, CA, USA) using acetonitrile and phosphoric acid (25 mM) as the mobile phases and an Agilent ZORBX RRHD SB C18 (2.1 × 50 mm, Agilent Technologies) chromatographic column at 40 °C, in which the injected volume was 20 μL, the flow rate was 0.5 mL/min, and the detector wavelength was fixed at 284 nm. For the calibration curve, standard solutions of 100, 200, and 300 ppm of 2,4-D were prepared using distilled water, and samples at 0, 10, 30, 60, and 120 min were taken and stored in amber vials. The adsorption capacity ($q_t$ mg/g) and removal percentage ($\mu$) of 2,4-D were calculated according to Equations (1) and (2), respectively:

$$q_t = \frac{V(C_i - C_f)}{m} \tag{1}$$

$$\mu = \frac{C_i - C_f}{C_i} \times 100 \tag{2}$$

where $V$ (L) is the solution volume, $C_i$ (mg/L) is the initial concentration of the solute, $C_f$ (mg/L) is the solute concentration in aqueous phase, and $m$ (g) is the mass of the adsorbent.

### 2.4.1. Effect of Solution pH

The effect of initial pH on the adsorption of 2,4-D using AC and MAC was studied by adjusting a solution of concentration 100 ppm. The initial solution pH was evaluated in the range of 2 to 10 at room temperature and was adjusted using 0.1 M HCl or 0.1 M NaOH solutions. The pH was measured using a pH meter (Orion Star A329, Thermo Scientific, Waltham, MA, USA). The equilibrium concentration was measured with a total contact time of 300 min.

### 2.4.2. Effect of Initial Concentration, Temperature, and Contact Time

Five initial concentrations (50, 100, 150, 200, and 300 ppm) of 2,4-D solution were investigated to measure the adsorption capacity of AC and MAC at different temperature (25, 35, and 45 °C); the pH was adjusted according to the results obtained previously. Samples were taken at 3, 5, 10, 30, 60, 180, 240, and 300 min.

Adsorption Kinetics

Adsorption rate values were obtained by adjusting the experimental data to four kinetic models: pseudo-first order, pseudo-second order, Elovich, and Webber and Morris equations (Equations (3)–(6), respectively) [31–33]:

$$\log(q_t - q_e) = \log q_e - \frac{k_1 t}{2.303} \tag{3}$$

$$\frac{t}{q_t} = \frac{1}{k_2 q_e^2} + \frac{t}{q_e} \tag{4}$$

$$q_t = \frac{1}{\beta} \ln \alpha\beta + \frac{1}{\beta} \ln t \tag{5}$$

$$q_t = k_p t^{1/2} + C \tag{6}$$

where $q_t$ (mg/g) is the solute adsorbed at the given time, $q_e$ (mg/g) is the solute adsorbed at the equilibrium, $k_1$ (1/min) is the pseudo-first-order constant, $t$ (min) is the given time, $k_2$ (g/mg·min) is the pseudo-second-order constant, $\alpha$ and $\beta$ are the Elvoich adsorption and desorption rate, respectively, and $k_p$ (mg/g·min$^{1/2}$) is the intraparticle diffusion constant.

### 2.4.3. Adsorption Isotherms

The adsorption data were adjusted using three isotherm models: Langmuir, Freundlich, and Elovich. The Langmuir model is based on the assumption that adsorption occurs on a homogeneous surface leading to the formation of a monolayer in which the adsorbed molecules do not interact with each other [34]. The mathematical model of Langmuir is given by Equation (7) [35]. Four linear forms of the Langmuir isotherm are shown in Table 1 [36].

$$q_e = \frac{q_m K_L C_e}{1 + K_L C_e} \tag{7}$$

**Table 1.** Lineal forms of Langmuir isotherms.

| Model | Lineal Form | |
|:---:|:---:|:---:|
| I | $\frac{C_e}{q_e} = \frac{1}{q_m K_L} + \frac{C_e}{q_m}$ | (8) |
| II | $\frac{1}{q_e} = \left[\frac{1}{q_m K_L}\right]\frac{1}{C_e} + \frac{1}{q_m}$ | (9) |
| III | $q_e = q_m K_L - \left[\frac{1}{K_L}\right]\frac{q_e}{C_e}$ | (10) |
| IV | $\frac{q_e}{C_e} = q_m K_L - q_e K_L$ | (11) |

The Freundlich model is based on the formation of a heterogeneous surface and is given by Equation (12) [37]. The Elovich model assumes an exponential increase of the adsorption on the surface, which is explained through Equation (13) [38].

$$q_e = K_f C_e^{\frac{1}{n}} \tag{12}$$

$$\frac{q_e}{q_m} = K_e C_e e^{-\frac{q_e}{q_m}} \tag{13}$$

Here, $q_e$ (mg/g) is the amount of solute adsorbed at the equilibrium, $q_m$ (mg/g) is the maximum solute adsorbed, $K_L$ is the Langmuir constant at the equilibrium, $C_e$ is the concentration of solute, $K_f$ and $n$ are the Freundlich constants, and $K_e$ is the Elovich constant.

## 2.5. Regeneration Study

Two regeneration methods were used to establish the efficiency of AC and MAC. Firstly, the chemical regeneration consisted of the addition of 1 g of AC or MAC in 100 mL of 0.1 M $HNO_3$ solution under constant agitation for 30 min. Subsequently, samples were washed with 200 mL of distilled water until the pH was neutralized. The second method was based on thermal regeneration, in which the AC and MAC were treated at from 25 to 400 °C for 20 min with a rate of increase of 5 °C/min. The samples were then washed three times with 100 mL of distilled water each. The efficiency of the regenerated carbons was evaluated by performing adsorption experiments using 2,4-D for five cycles.

## 3. Results and Discussion

### 3.1. Characterization of AC and MAC

#### 3.1.1. SEM-EDS

The morphological and elemental information of the carbon (C), activated carbon (AC), and magnetic activated carbon (MAC) are shown in the SEM images and EDS spectra in Figure 1. The carbon surface exhibited porosity with pores of different sizes and shapes, which is suitable for the adsorption of 2,4-D. In Figure 1a it can be observed that the C had a uniform distribution of pores with a smooth surface. In Figure 1b, a porous and rough surface was observed for AC, as well as some notches in each section of the surface, and porosity with low uniformity was evidenced. Porosity in the structure of AC can be strongly attributed to activation with KOH; similar results have been reported in the literature [6]. The SEM image in Figure 1c corresponds to the MAC and evidences a decrease in porosity compared to the AC. The MAC presents a rough surface with low uniformity due to modification with $Fe_3O_4$ nanoparticles which may be occupying the pores and thus reducing the possible active sites for the adsorption of 2,4-D.

Figure 1d shows the EDS results for AC and MAC. In the spectra are evidenced the presence of potassium as result of the chemical activation using KOH. Iron and oxygen elements in the MAC indicate successful modification with $Fe_3O_4$ nanoparticles. Both carbons contain some traces of phosphorus due to chemical treatment with $H_3PO_4$. Figure 1e shows the element mapping for a specific area of the MAC's surface which evidences the distribution of iron. The distribution indicates that the $Fe_3O_4$ nanoparticles present uniformity on the MAC's surface, which reduces the porosity.

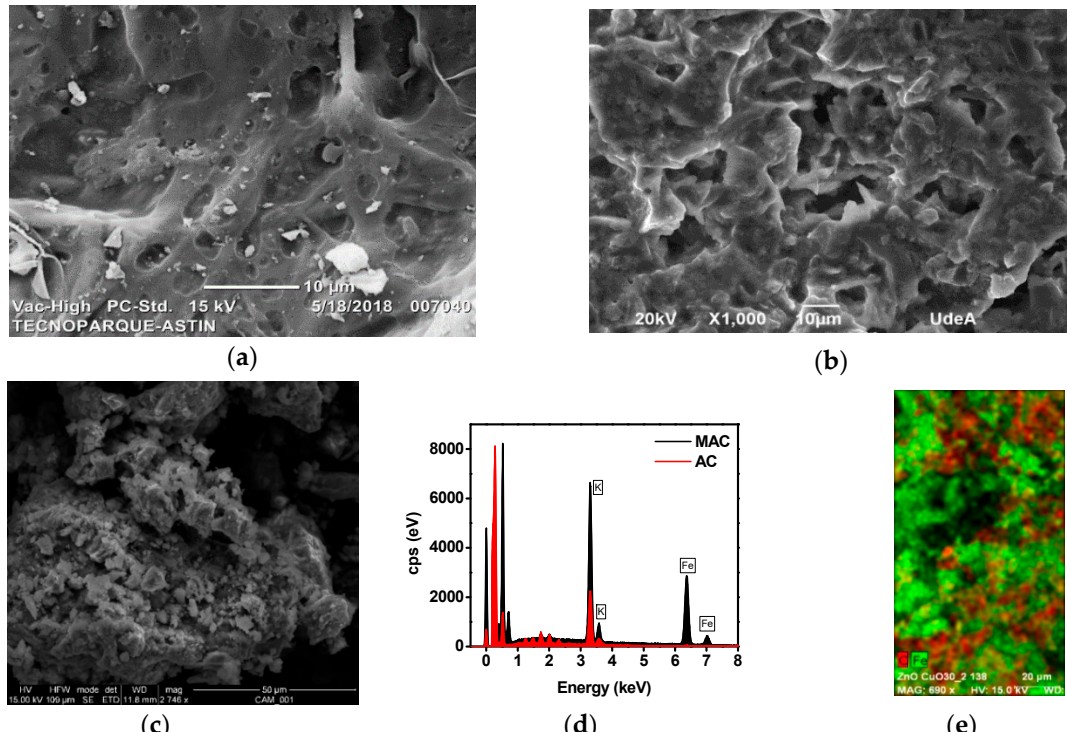

**Figure 1.** SEM images of (**a**) C, (**b**) activated carbon (AC), (**c**) and magnetic AC (MAC), (**d**) EDS spectra and (**e**) mapping.

### 3.1.2. BET (Brunauer, Emmett and Teller) Surface Area

BET analysis was performed for the AC and MAC. The results indicate that the highest surface area was 715.12 $m^2$/g for AC, which means an improvement compared to the C with a value of 506.18 $m^2$/g. The surface area improvement may be attributed to the chemical activation process, as was reported by similar studies using KOH for the activation of carbons [39]. In case of MAC, the surface area decreased to 325.22 $m^2$/g, which is in agreement with the results showed in the SEM images in Figure 1. However, the results on the surface area for AC and MAC are comparable with other results reported in the literature in which different agroindustrial wastes were used, such as langsat empty fruit carbon (1065.65 $m^2$/g) [18] and groundnut shell carbon (709 $m^2$/g) [6]. The total pore volume was also obtained through BET analysis, indicating that AC and MAC present a microporous structure with volumes of 0.300425 $cm^3$/g and 0.141363 $cm^3$/g, respectively.

### 3.1.3. XRD

The XRD patterns of the $Fe_3O_4$ nanoparticles, AC, and MAC are shown in Figure 2a. In accordance with the standard card JCPDS No.65-3107, the pattern for $Fe_3O_4$ nanoparticles showed the characteristic peaks at 30.2, 35.7, 43.1, 53.8, 57.2, and 62.9° which correspond to the (220), (311), (400), (422), (511), and (440) crystal planes, associated with a cubic structure [26,27,40].

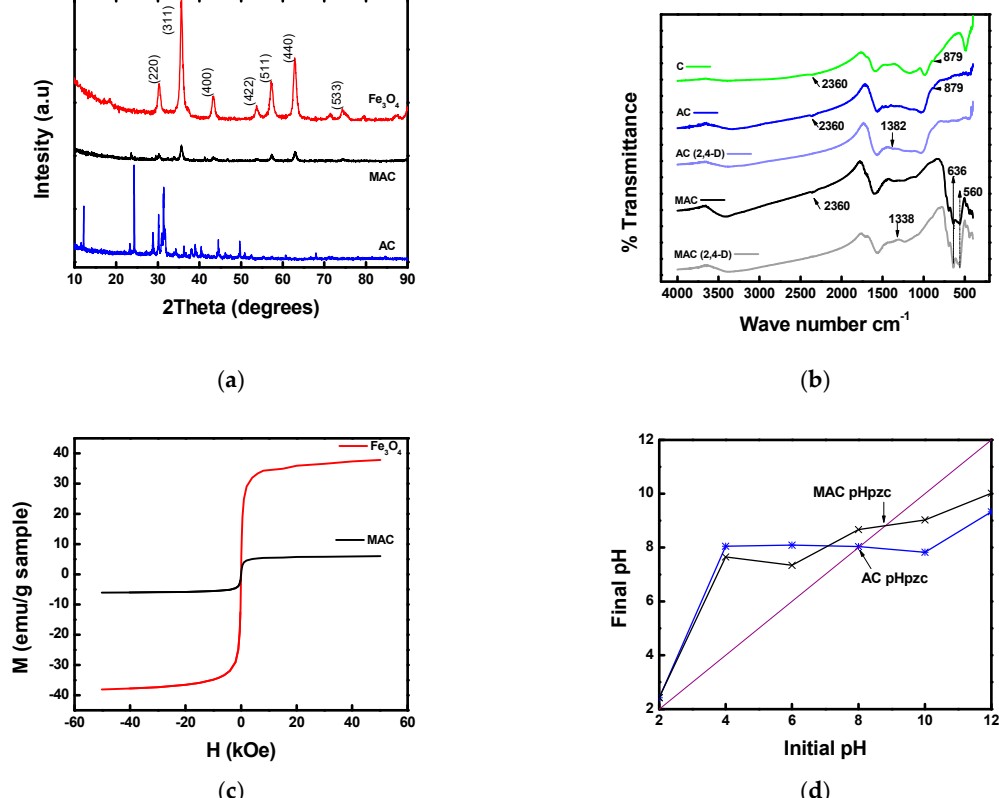

**Figure 2.** (**a**) XRD patterns of samples; (**b**) FTIR spectra of samples, 2,4-dichlorophenoxyacetic acid (2,4-D) after adsorption of pesticide; (**c**) Hysteresis loop for MAC and $Fe_3O_4$; (**d**) Determination of the point of zero charge (pHpzc).

From the pattern for AC it was possible to observe a natural amorphous structure with two characteristic peaks at 25° and 30°, and according to the pattern for MAC, the same characteristic peaks for the carbon structure were observed, as well as the peaks corresponding to the structure of the $Fe_3O_4$ nanoparticles [41].

### 3.1.4. FTIR

The functional groups present in the C, AC, and MAC were studied using FTIR spectroscopy as shown in Figure 2b. The IR spectra describe the active chemical sites on the surface of the carbons, in which the adsorption of 2,4-D may be carried out. The active sites of the AC and MAC presented low intensity compared to those of the C; some peaks disappeared, but two stretching vibrations were also observed as the most representative peaks at 879 and 2360 $cm^{-1}$. These results are in accordance with the carbon activation using KOH; therefore, the breaking of bonds may have caused the new stretching vibrations to arise [39]. Additionally, carbonization at high temperature may produce the breaking of bonds. The spectrum of MAC confirms that the carbons were treated and modified with $HNO_3$ and $Fe_3O_4$ nanoparticles, respectively [22]. AC and MAC present similar stretching vibrations at 650, 879, 1031, 1427, 1548, and 2360 $cm^{-1}$, which were assigned to carboxylic acids, C=C bonds in aromatic rings; C–O bonds present in alcohols, phenols, acids, esters, or ethers; and C≡C in alkaline groups. The MAC exhibited unique peaks at 560 and 636 $cm^{-1}$ which may be attributed to the presence of $Fe_3O_4$ in the structure [42].

After the adsorption of 2,4-D, samples of AC and MAC were taken to be analyzed using the FTIR technique. The spectra evidenced changes in peaks at 2360 $cm^{-1}$, which were attributed to the adsorption of 2,4-D. The shifting in the peak around 1300 $cm^{-1}$ confirms the adsorption of 2,4-D onto the surface, during which a donation of electrons by the molecule to the surface occurred [6].

### 3.1.5. VSM

The magnetization curve is shown in Figure 2c. The characterization was carried out at room temperature in a cycle between −50 KOe and 50 KOe. According to the curve, the magnetization saturation (Ms) values were 6 emu/g and 38 emu/g for MAC and $Fe_3O_4$ nanoparticles, respectively. Typical superparamagnetic behavior was observed for both materials and evidenced by the shape of the characteristic curve [43]. The decrease in the magnetic response for MAC is evident when compared to the nanoparticles; this is attributed to the amount of magnetic material contained in the carbon structure. As a result of the magnetic response of MAC, it was possible to separate the material by applying a magnetic field after the adsorption experiments.

### 3.1.6. Point of Zero Charge (pH$_{pzc}$)

In Figure 2d it can be observed that the pH$_{pzc}$ values were 8 and 8.9 for AC and MAC, respectively. According to these results, both carbons may have affinity toward anionic molecules when the pH value is below the obtained pH$_{pzc}$. This is due to the increase of cations on the surface of the carbons. In the case of a pH value greater than pH$_{pzc}$, the adsorption might be favorable for cationic compounds [30]. 2,4-D disassociates in water, forming anionic molecules; hence, the as-prepared carbons are expected to work as potential adsorbents of this pollutant at pH below pH$_{pzc}$.

### 3.2. Adsorption Study

### 3.2.1. Effect of Contact Time, Initial Concentration, and Temperature

Figure 3 shows the effect of contact time on the adsorption of 2,4-D ($q_t$) onto AC and MAC at 25 °C using initial concentrations of 50, 100, 150, 200, and 300 ppm. In Figure 3a, the equilibrium for adsorption onto the AC´s surface was achieved after 240 min for all initial concentrations. In the case of MAC, equilibrium was reached after 150 min also for all initial concentrations as shown in Figure 3b. These results suggest that the surfaces of AC and MAC have a great number of active sites, which is in agreement with the information obtained from the analysis of SEM images and BET previously. Therefore, the saturation of the active sites on the surface and the equilibrium occur rapidly. The adsorption capacities were calculated to be 29.36 and 32.32 mg/g for the MAC and AC, respectively, after 3 min when the initial concentration was 300 ppm. This result indicates that the driving forces (concentration gradients) are significant during the adsorption process in the first 50 min, promoting high and fast adsorption rates [7].

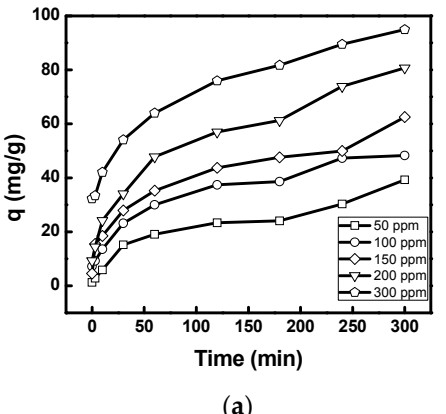
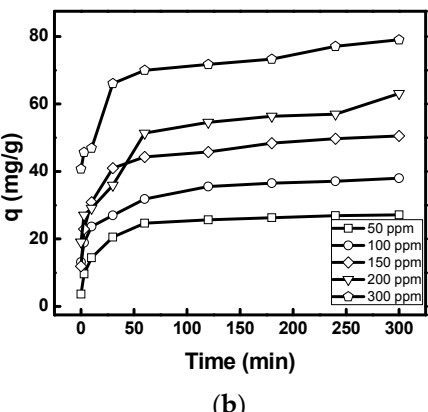

|     |     |
| :-: | :-: |
| (**a**) | (**b**) |

**Figure 3.** Effect of contact time on the adsorption capacity at 25 °C on (**a**) AC and (**b**) MAC.

In order to corroborate the results of 2,4-D adsorption onto AC and MAC, a kinetic adsorption curve was obtained at 100 mg/L and 25 °C using HPLC coupled with a UV–vis detector. Figure 4a shows a comparison between the UV–vis and HPLC–UV results. The kinetic curve obtained using

HPLC–UV showed a slight variation with respect to that using UV–vis. An average of 6.19% (4.67 mg/L) was noted for five points in the curve. Therefore, the measurement of the 2,4-D concentration using UV–vis was appropriate, allowing us to obtain similar values around 2.36 mg/L.

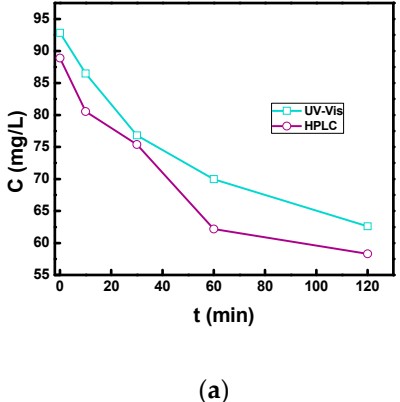

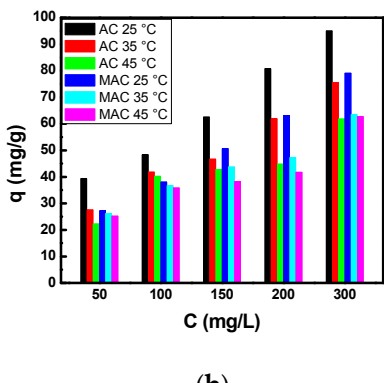

(**a**)                                                                              (**b**)

**Figure 4.** (**a**) Comparison between UV–vis and HPLC techniques in the adsorption of 2,4-D in AC at 25 °C; (**b**) Effect of initial concentration and temperature on the adsorption capacity.

In Figure 4b, both adsorbents showed high adsorption capacity as the initial concentration increased. For AC and MAC, the $q_e$ values increased from 39.25 to 94.96 mg/g and from 27.14 to 79.05 mg/g, respectively. Accordingly, at low initial concentration, the abundance of active sites on the surface promotes an improvement in the adsorption, indicating that the adsorption is highly dependent on the initial concentration. The negative effect of temperature on the removal of 2,4-D can be observed in Figure 4b, which is attributed to an increase in 2,4-D solubility leading to desorption [44]. This result can also be explained by the fact that 2,4-D molecules adsorbed onto the surface have a strong vibrational energy that overcomes the surface forces, allowing spread to the liquid phase [45]. Diaz-Flores et al. reported negative effects of temperature on the adsorption capacity of activated carbon [39].

### 3.2.2. Effect of pH

Figure 5 shows the results of the adsorption capacity ($q_e$) of 2,4-D on AC and MAC at different pH. For pH 2, the adsorption capacities were found to be 93.35 and 86.74 mg/g for AC and MAC, respectively. The trend of the plot shows a plateau between pH 4 and 8 for the adsorption capacity of AC, which is the opposite of the tendency for MAC. These results are explained by the electrostatic interaction between charges of the 2,4-D molecule and the adsorbent surface [46]. The dissociation of anionic compounds in aqueous solutions is propitiated by pH values higher than the acid dissociation constant (pKa). In the case of 2,4-D, the value was 2.8, confirming that it behaves as an anionic compound with increasing pH [47]. Therefore, a decrease in the adsorption capacity on the surface is mainly due to repulsion by negative charges in the medium. During the acidification of the medium, protonation of the adsorbent surface occurs, increasing the electrostatic interaction towards the herbicide and thus increasing the adsorption capacity.

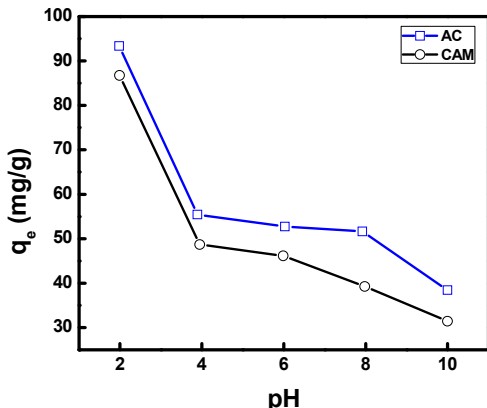

**Figure 5.** Effect of pH on the adsorption capacity at 25 °C and a 2,4-D initial concentration of 100 mg /L.

### 3.2.3. Kinetic Study

The experimental data were fitted to kinetic and intraparticle diffusion models. From Figure 6a,b and Tables 2 and 3, the 2,4-D adsorption experimental data fitted the three kinetic models, among which the pseudo-second-order model showed the best coefficient ($R^2$) average value. For adsorption at 25 °C, $R^2$ average values of 0.9921 and 0.9974 were obtained for AC and MAC, respectively. These results suggest that limiting of the adsorption mechanism was mainly due to chemical interactions (chemisorption) that involve valence and electron exchange forces between the adsorbent and adsorbate [48,49]. In the literature it is reported that the pseudo-second-order model better predicts the adsorption of 2,4-D onto carbon's surface, as well as that of pesticides as adsorbates such as ametryn, aldicarb, dinoseb, diuron [50], and carbofuran [5]. The equilibrium adsorption capacity ($q_e$) calculated using this model showed a similar value to that calculated from the experimental data, obtaining an average percentage of 95.27 with a standard deviation of ±2.2391 mg/g.

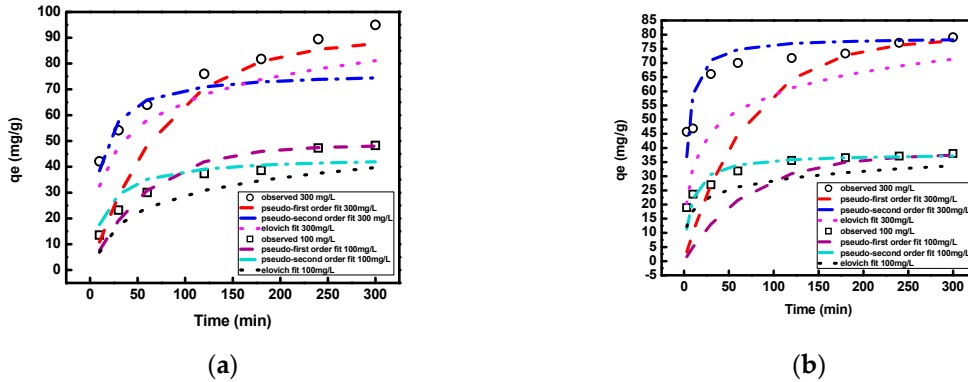

(**a**)        (**b**)

**Figure 6.** Adjustment to the kinetic models for the adsorption of 2,4-D on (**a**) AC and (**b**) MAC at 100–300 mg/L and 25 °C.

**Table 2.** Kinetic parameters of the models studied for 2,4-D on AC at different initial concentrations and 25 °C.

| $C_o$ (mg/L) | $q_{e,exp}$ (mg/g) | Pseudo-First Order | | | Pseudo-Second Order | | |
|---|---|---|---|---|---|---|---|
| | | $K_1$ (1/min) | $q_e$ | $R^2$ | $K_2$ (mg/g min) | $q_e$ | $R^2$ |
| 50 | 27.50 | 0.012 | 27.51 | 0.94 | 0.0010 | 29.85 | 0.98 |
| 100 | 41.78 | 0.017 | 41.79 | 0.80 | 0.0015 | 44.05 | 0.99 |
| 150 | 46.63 | 0.013 | 46.64 | 0.97 | 0.0013 | 48.31 | 0.99 |
| 200 | 61.88 | 0.015 | 61.89 | 0.94 | 0.0008 | 65.36 | 0.99 |
| 300 | 75.51 | 0.013 | 75.50 | 0.99 | 0.0013 | 76.92 | 0.99 |

| $C_o$ (mg/L) | $q_{e,exp}$ (mg/g) | Elovich | | | Intraparticular Diffusion Model | | |
|---|---|---|---|---|---|---|---|
| | | $\beta$ (g/mg) | $\alpha$ (mg/g min) | $R^2$ | $K_p$ (mg/g min$^{1/2}$) | $C_i$ | $R^2$ |
| 50 | 27.5 | 0.193 | 0.382 | 0.96 | 1.664 | 1.651 | 0.94 |
| 100 | 41.78 | 0.144 | 1.382 | 0.96 | 2.230 | 9.085 | 0.93 |
| 150 | 46.63 | 0.121 | 0.914 | 0.99 | 2.565 | 8.168 | 0.91 |
| 200 | 61.88 | 0.093 | 0.983 | 0.97 | 3.430 | 10.391 | 0.94 |
| 300 | 75.51 | 0.107 | 9.739 | 0.97 | 3.009 | 29.646 | 0.95 |

**Table 3.** Kinetic parameters of the models studied for 2,4-D on MAC at different initial concentrations and 25 °C.

| $C_o$ (mg/L) | $q_{e,exp}$ (mg/g) | Pseudo-First Order | | | Pseudo-Second Order | | |
|---|---|---|---|---|---|---|---|
| | | $K_1$ (1/min) | $q_e$ | $R^2$ | $K_2$ (mg/g min) | $q_e$ | $R^2$ |
| 50 | 27.14 | 0.012 | 27.143 | 0.78 | 0.0054 | 27.55 | 0.99 |
| 100 | 37.99 | 0.011 | 37.991 | 0.92 | 0.0037 | 38.31 | 0.99 |
| 150 | 50.55 | 0.012 | 50.552 | 0.92 | 0.0032 | 50.76 | 0.99 |
| 200 | 63.11 | 0.001 | 63.113 | 0.91 | 0.0014 | 62.11 | 0.99 |
| 300 | 79.05 | 0.011 | 79.058 | 0.94 | 0.0022 | 78.74 | 0.99 |

| $C_o$ (mg/L) | $q_{e,exp}$ (mg/g) | Elovich | | | Intraparticular Diffusion Model | | |
|---|---|---|---|---|---|---|---|
| | | $\beta$ (g/mg) | $\alpha$ (mg/g min) | $R^2$ | $K_p$ (mg/g min$^{1/2}$) | $C_i$ | $R^2$ |
| 50 | 27.14 | 0.2381 | 2.99 | 0.98 | 1.245 | 9.47 | 0.81 |
| 100 | 37.99 | 0.2281 | 21.36 | 0.99 | 1.357 | 17.73 | 0.90 |
| 150 | 50.55 | 0.1519 | 9.48 | 0.98 | 1.953 | 22.03 | 0.81 |
| 200 | 63.11 | 0.1304 | 8.38 | 0.95 | 2.464 | 23.11 | 0.93 |
| 300 | 79.05 | 0.1411 | 21.78 | 0.95 | 2.207 | 44.94 | 0.87 |

The Elovich model showed average $R^2$ values of 0.9725 and 0.9682 for AC and MAC, in which inverse and direct proportional relationships were observed for the $\beta$ and $\alpha$ constants, respectively. For AC, the $\alpha$ value is equivalent to the initial concentration gradient and varies between 50 and 100 ppm with values from 0.3816 to 9.7393 mg/g min. The values obtained for $\beta$ decreased from 0.1932 to 0.1067 g/mg and from 0.2381 to 0.1114 g/mg with increasing initial concentration for the AC and MAC, respectively. These results can be explained by the fact that the adsorption mechanism was mainly due to chemical interactions [51], which was also evidenced in the results using the pseudo-second-order model.

The pseudo-first-order and intraparticle diffusion models showed $R^2$ average values below 0.93. Although the prediction of the data was good for the results using AC and MAC, the pseudo-second-order and Elovich models proved to be better. In Figure 7a,b can be initially observed linearly increasing behavior of $q_t$ versus $t^{1/2}$ for both carbons in the first 60 min (7.74 min$^{1/2}$). Afterwards, the linear behavior decreased until the equilibrium was reached at 300 min. This result is in agreement with Fierro et al., who explained that each stage during the multilinear process is related initially with macroporous diffusion and then in the second instance with microporous diffusion [52].

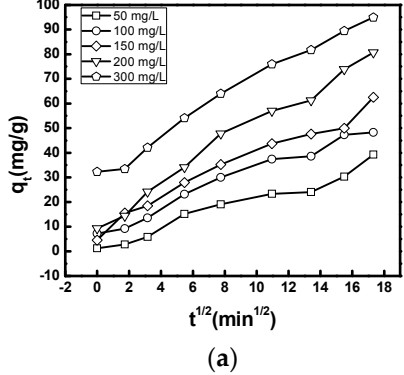 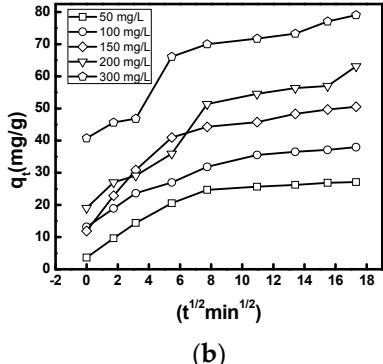

(**a**)        (**b**)

**Figure 7.** Adjusted intraparticle diffusion model for the adsorption of 2,4-D onto (**a**) AC and (**b**) MAC at different initial concentrations and 25 °C.

### 3.2.4. Adsorption Isotherms

Langmuir (type I, II, III, and IV according to Table 1), Freundlich, and Elovich models were used in order to study the interaction between the carbons and 2,4-D in solid and liquid phase during the adsorption experiments. In Table 4 are included the equation parameters for all models. Figure 8a,b shows the curves when the experimental data were fitted to the Freundlich and Langmuir isotherm models. From the $R^2$ values it can be observed that the Freundlich model better fit the experimental data, obtaining average values of 0.9330 and 0.9568 for AC and MAC, respectively. Moreover, this model showed that the values of the $n_f$ coefficient are greater than the unit at all temperatures, indicating good adsorption in the system. However, a decrease in this constant was observed relative to the temperature for both carbons.

**Table 4.** Parameters of the different isothermal models for AC and MAC.

| Isotherm Model | | AC | | | MAC | | |
|---|---|---|---|---|---|---|---|
| | T (°C) | 25 | 35 | 45 | 25 | 35 | 45 |
| Langmuir | $K_L$ (L/mg) | 0.0382 | 0.0266 | 0.0291 | 0.0223 | 0.0309 | 0.0281 |
| | $q_m$ (mg/g) | 99.0099 | 80.0000 | 63.2911 | 86.2069 | 64.5161 | 60.9756 |
| | $R^2$ | 0.9213 | 0.9042 | 0.9187 | 0.8801 | 0.9279 | 0.8755 |
| Freundlich | $K_f$ (mg/g)/(mg/L)$^{1/n}$ | 1.3067 | 1.1202 | 1.0970 | 1.1021 | 1.1257 | 1.2108 |
| | $n_f$ | 1.3144 | 1.2469 | 1.2341 | 1.3519 | 1.3215 | 1.2296 |
| | $R^2$ | 0.8761 | 0.9596 | 0.9633 | 0.9694 | 0.9518 | 0.9493 |
| Elovich | $K_E$ | 0.1801 | 0.0513 | 0.0491 | 0.0333 | 0.0930 | 0.0550 |
| | $q_m$ (mg/g) | 34.6021 | 39.6825 | 32.6797 | 49.0196 | 26.1780 | 29.9401 |
| | $R^2$ | 0.6927 | 0.8477 | 0.7408 | 0.8469 | 0.8672 | 0.6321 |
| Langmuir II | $K_{LII}$ (L/mg) | 0.0936 | 0.0251 | 0.0162 | 0.0221 | 0.0316 | 0.0342 |
| | $q_m$ (mg/g) | 75.7576 | 74.6269 | 74.6269 | 78.1250 | 59.5238 | 53.7634 |
| | $R^2$ | 0.7309 | 0.9426 | 0.9519 | 0.9197 | 0.9371 | 0.8743 |
| Langmuir III | $K_{LIII}$ (L/mg) | 0.0808 | 0.0221 | 0.0199 | 0.0189 | 0.0281 | 0.0304 |
| | $q_m$ (mg/g) | 81.0900 | 79.4410 | 66.7290 | 84.6980 | 62.3777 | 56.7530 |
| | $R^2$ | 0.5192 | 0.7622 | 0.7600 | 0.7192 | 0.7625 | 0.5790 |
| Langmuir IV | $K_{LIV}$ (L/mg) | 0.0420 | 0.0168 | 0.0147 | 0.0136 | 0.0214 | 0.0176 |
| | $q_m$ (mg/g) | 95.7786 | 88.5714 | 75.3605 | 97.4853 | 67.9206 | 68.4943 |
| | $R^2$ | 0.5192 | 0.7622 | 0.0147 | 0.7192 | 0.7625 | 0.5790 |

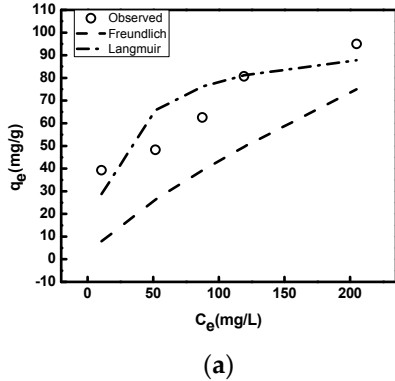 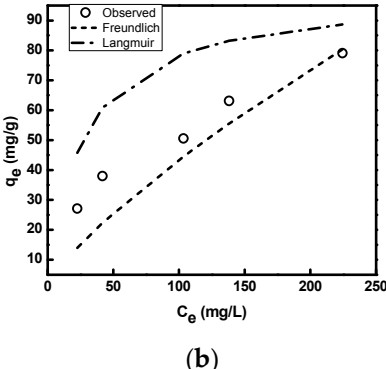

| (a) | (b) |

**Figure 8.** Nonlinear Freundlich and Langmuir isotherms for 2,4-D adsorption onto (**a**) AC and (**b**) MAC at 25 °C.

The $K_f$ constant is considered the adsorption index of the adsorbent material; the value decreased from 1.3067 to 1.0970 as the temperature increased. This result indicates that the adsorption process has an exothermic nature, defining the increase in the temperature as negative, such as the value obtained at 45 °C which was close to the unit [53].

The $R^2$ average values for the Langmuir model were 0.9147 and 0.8945 for the AC and MAC, respectively. The Langmuir model separation constants ($R_L$) were found to be within the range between 0 and 1, indicating that the adsorption process was favorable [5]. The maximum adsorption capacity values ($q_m$) at 25 °C were 99.009 and 86.207 mg/g for AC and MAC, respectively. These values are in agreement with those reported in the literature for the adsorption of 2,4-D using surpassing coals and activated carbons made with corn residues (95.26 mg/g), rice (1.4 mg/g), bagasse ash (3.82 mg/g), filter paper and cotton (77.3 mg/g), and peanut peel (3.02 mg/g) [7].

According to the results using Freundlich and Langmuir models, the adsorption process was carried out mainly by chemisorption leading to the formation of a monolayer, taking into account that the Freundlich model is normally associated with the formation of multilayers [54].

### 3.2.5. Thermodynamic Study

In order to study the thermodynamic nature of the 2,4-D adsorption onto AC and MAC, enthalpy changes ($\Delta H$) were evaluated to determine if the process is proportional to incoming or outgoing energies. Moreover, Gibbs free energy ($\Delta G$), which explains the reversibility, and entropy ($\Delta S$), which relates the randomness in thermodynamic processes, were also evaluated. The Clausius–Clapeyron equation can be linearized as follows:

$$\ln(C_e) = \frac{\Delta H}{RT} + C \tag{14}$$

where $C_e$ (mg/L) is the 2,4-D equilibrium concentration in solution, $R$ (J/mol K) is the universal gas constant, and T (K) is the temperature. According to the adsorption results using the Freundlich model, $\Delta G$ values were calculated using an equation reported by Huang et al. [55]. $\Delta S$ was calculated using a thermodynamic relation.

$$\Delta G = -n_f RT \tag{15}$$

$$\Delta S = \frac{\Delta H - \Delta G}{T} \tag{16}$$

In Tables 5–7 are presented the results for $\Delta H$, $\Delta G$, and $\Delta S$, respectively. The initial concentration of 2,4-D and temperature were adjusted to 100, 200, and 300 ppm and to 25, 35, and 45 °C accordingly. Then, $R^2$ average values were calculated for $\Delta H$ as shown in Table 5, in which the data fitted the

Clausius–Clapeyron linearized equation properly, obtaining values of 0.9217 and 0.9017 for AC and MAC, respectively. Although the experimental data fitted the equation, negative values were calculated, confirming that the adsorption of 2,4-D onto the as-prepared adsorbents had an exothermic nature. In the case of $\Delta G$ in Table 6, negative values determined that the adsorption process presented a spontaneous nature, which can be explained by considering the high surface area of the carbons as an advantage during the adsorption process. Additionally, the $\Delta S$ values observed in Table 7 indicate a tendency of increasing as the temperature increases; similar behavior was evidenced by $\Delta G$ values. These negative values are related with a lower adsorption capacity due to the occurrence of minor changes in the adsorbate/adsorbent interaction [21].

**Table 5.** Enthalpy change at 100, 200, and 300 ppm for AC and MAC.

|  | **AC** | | | **MAC** | | |
|---|---|---|---|---|---|---|
| $C_o$ (mg/L) | 100 | 200 | 300 | 100 | 200 | 300 |
| $\Delta H$ (J/mol) | −6837.49 | −10.402.47 | −712.28 | −1379.12 | −5760.18 | −2849.45 |
| $R^2$ | 0.7775 | 0.9979 | 0.9897 | 0.9872 | 0.9293 | 0.7977 |

**Table 6.** Gibbs free energy change at 25, 35, and 45 °C for AC and MAC.

|  | **AC** | | | **MAC** | | |
|---|---|---|---|---|---|---|
| $T$ (°C) | 25 | 35 | 45 | 25 | 35 | 45 |
| $\Delta G$ (J/mol) | −491.75 | −362.83 | −256.51 | −505.78 | −384.55 | −255.56 |

**Table 7.** Entropy change at 100, 200, and 300 ppm at 25, 35, and 45 °C for AC and MAC.

| | $\Delta S$ (J/mol K) | | | | | |
|---|---|---|---|---|---|---|
| **$C_o$ (mg/L)** | **AC** | | | **MAC** | | |
| | **25 (°C)** | **35 (°C)** | **45 (°C)** | **25 (°C)** | **35 (°C)** | **45 (°C)** |
| 100 | −21.2587 | −20.9876 | −20.6624 | −2.9258 | −3.1227 | −3.2239 |
| 200 | −33.2017 | −32.5434 | −31.8555 | −16.8780 | −17.4251 | −17.6027 |
| 300 | −0.7388 | −1.1327 | −1.4310 | −7.8515 | −7.9900 | −7.991 |

### 3.2.6. Regeneration Capacity of AC and MAC

After the chemical regeneration process, MAC showed higher adsorption capacity compared to AC, as shown in Figure 9a. After the first regeneration cycle, a 55% loss of adsorption capacity was evidenced, whereas the MAC presented 91.92% efficiency after the fifth cycle. The regeneration capacity of MAC may be attributed to the presence of $Fe_3O_4$ nanoparticles in the structure. This result is in agreement with a similar study using magnetic nanoparticles within the porous carbon structure, contributing to the regeneration capacity and preventing deactivation effects [10].

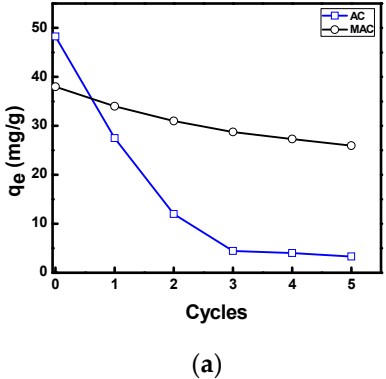 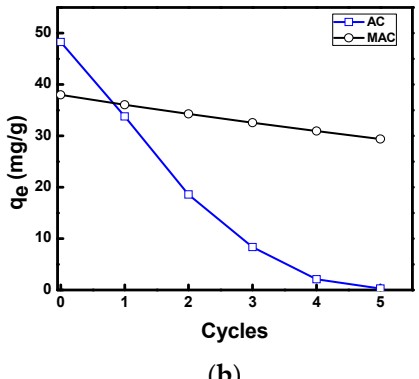

(**a**)　　　　　　　　　　　　　　　　　　　　　　(**b**)

**Figure 9.** Equilibrium adsorption of 2,4-D onto AC and MAC adsorbents after (**a**) chemical and (**b**) thermal regeneration processes.

As shown in Figure 9b, thermal regeneration decreased the adsorption capacity to 70% after the first cycle, indicating positive effects compared to the chemical process. However, the loss increased up to 82.6% after the third cycle, and then a complete loss of 100% was achieved after the fifth cycle. These results can be attributed to the formation and accumulation of adsorbent residues due to decomposition of the carbon at the regeneration temperature, which was relatively high for organic compounds [11].

## 4. Conclusions

In this investigation we reported the successful synthesis of MAC from *D. rotundata* peels for the removal of 2,4-D from aqueous solution. An improvement in the surface area of the AC was observed due to the chemical activation of C with KOH. $Fe_3O_4$ nanoparticles synthesized by a coprecipitation method were obtained, which allowed us to confer superparamagnetic properties to the AC. The results revealed that the adsorption capacity of 2,4-D onto AC and MAC was enhanced with increasing contact time and initial concentration, but it was also reduced with increasing temperature and solution pH. The adsorption kinetics and isotherms were better described through the pseudo-second-order and Freundlich models, respectively. The thermodynamic parameters showed that the energetic nature of 2,4-D adsorption onto the AC and MAC was that of an exothermic, spontaneous, and entropic process. MAC presented a better regeneration capacity than AC after five cycles. We also concluded that MAC could be separated easily without performing an additional filtration step. AC and MAC can be obtained from agroindustrial residues of *D. rotundata* and used as efficient alternatives for the adsorption and separation of organic pollutants in aqueous solution.

**Author Contributions:** Conceptualization, U.H.-G., J.C., D.P.-R., R.S. and A.H.; data curation, U.H.-G.; formal analysis, U.H.-G., J.C. and R.S.; funding acquisition, A.H.; investigation, U.H.-G., D.P.-R. and R.S.; methodology, U.H.-G., J.C., R.S., D.P.-R. and A.H.; project administration, A.H.; resources, A.H.; supervision, A.H.; validation, A.H.; writing—original draft, U.H.-G.; writing—review and editing, U.H.-G., D.P.-R., R.S. and A.H.

**Funding:** This research was funded by Universidad de Cartagena, with support from the project grants No. 106-2017 and No. 062-2018.

**Acknowledgments:** The authors would like to acknowledge Universidad de Cartagena for the financial support. Furthermore, the authors are grateful to Bianchi Mendez from the Electronic Nanomaterials Physics Research group at Universidad Complutense de Madrid (Spain) for her contribution with technical support.

**Conflicts of Interest:** The authors declare no conflict of interest.

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
