# Peer review of "Activated Carbon from Yam Peels Modified with Fe3O4 for Removal of 2,4-Dichlorophenoxyacetic Acid in Aqueous Solution"

_water, doi:10.3390/w11112342_

Round 1
Reviewer 1 Report
The current study is an attempt to assess the adsorption efficacy of the pesticide 2,4-dichlorophenoxyacetic acid (2,4-D) using activated carbon modified with Magnetite (Fe3O4) nanoparticles in an aqueous solution. The manuscript holds severe laps in writing style/typos. Moreover, the scientific soundness is very low. Therefore, it is suggested to undergo major revisions in the submitted manuscript.
Line 21! Word maximums should be changed to maximum.
Abstract needs to be rewritten. Looks more descriptive explanation of materials and methods rather than summarizing results.
Why ADRWC/Fe3O4 showed less adsorption capacity than ADRWC. If so, why do you synthesize ADRWC/Fe3O4?
Line 29-46 First three paragraphs of introduction should be merged together. Moreover, writing style should be scientific. it looks more like report. Revise accordingly. Describe literature reported values of 2,4D in different water bodies of the world.
There should be some relevant literature to adsorption capacities of activated carbon towards 2, 4 D. Must add.
Line 55-63! Generic detail. Summarize in 2-3 lines.
Line 70-78. Describe aims and objectives of study rather than methodology.
Line 87! What is INVESA Colombia?
Line 92-92! What was NaOH addition rate in mL/min?
Line 103-104! During carbonization, how do you provide airfree environment/anaerobic conditions?
Line 98-109! Looks confusing. Revise!
Line 128-131! Placing references and don't providing details of methodology is not appropriate way of describing methodology. Write full details, how point of zero charge was determined experimentally.
Line 136! 50mL 2,4-D solution contains how much 2, D in ppm? Write values.
Line 155! Initial solution pH correct it.
Figure 1(d)! It is obvious that authors have replace the EDS spectra data of ADRWC with ADRWC/Fe3O4? Big mistake correct it.
Overall, presentation of data in graphs is not appropriate. Why authors represents all graphs in the of concentration why not adsorption capacities. Revise all graphs and report results accordingly.
Where is statistical models. The models used are isotherms, kinetics and thermodynamics not statistical. The word statistic sounds some p value and others?
The authors must revise all the manuscript. It needs severe revision, otherwise it will be rejected for publications.
Reviewer 2 Report
Dear authors,
Thank you for an interesting article on how to use prosed Yam to remove 2,4-dichlorophenoxyacetic from water. I fully appreciate the aim of what you want to do.
I only have one main objection and that is that you need to be much more precise on how to do things. You have to write it so that someone else can follow it! Things which work a one scale may not work so well at another. For examples where I think you need to med more precise:
L89/96) How much of the different materials goes into Magnetite Nanoparticles?
Not “washed several” time but washed with ???ml water minimum X time and with ??? mL EtOH minimum Y time.
L97/109) How much of the different materials goes into Magnetite Nanoparticles?
Is it correct” The size of dry material was reduced to 200 mm size. That’s big? How big are the Yam, waste? (A picture or a drawing would be nice)
L111) What is the “in definite amounts” write a number
L114) Not “several times” but minimum number or from 4 to 6 times, not several times
L115 Dried at 80C for how long?
L155) Do you mean that the pH the initial solution was adjusted? To what? If not, maybe you could write something like: The pH of the initial solution was adjusted with HCl (0. M) and /or NaOH8 (0.1) to be in the range of 2-10?
L161 When is a convenient time? Write when
Other minor suggestions:
L166/171, Could you move the explanation for what letters in the equations stand for up so that they are just below the equations, where they are first mentioned
L174) Think k2 is g/(mg min) not (g/mg min1/2)”
L191-199) You forgot to lower the numbers in Fe3O4 and HNO3
Amount of chemicals used, please state
Still, do not like “washed several time” it is too unprecise
I don’t think you should use an abbreviation in the title, use 2,4-dichlorophenoxyacetic acid instead
L39) Please have a look at this sentence “After spraying, a large percentage of pesticides end up in bodies of water and rivers due to natural factors such as the carryover of these by rainfall, contaminating water sources”
L218 and 223) Fe3O4
L237) m2/g
L272) What does “cycle from 5 KOe” mean
L295) Think you are missing a “not” in “was reached”
Round 2
Reviewer 1 Report
Unsatisfactory. The manuscript must be rejected since author fails to address the comments.
When I go through the revised version of the manuscript, almost all major comments were not addressed in a scientific way, as it should be for any scientific paper. I am giving more details as below:
1) The abstract looks like reporting the result. There is no introduction line, no information regarding the main water chemistry parameters. For example, at what pH and if the pHpzc of adsorbent is 8 or 8.9 it means it is more favorable for anionic sorbent but authors use activated carbon modified activated carbon without any acidic treatment, how pHpzc become 8 or 8.9, Just using iron salt can't let this happen. Not correctly reported results or at least incomplete information. Other information like if authors are conducting experiments under certain conditions, they should provide the main finding rather than enlisting all results. Finally, in the abstract, concluding remarks should be more persuasive, more clear remarks towards achievement or new findings in the research field.
2) Point 3: Why ADRWC/Fe3O4 showed less adsorption capacity than ADRWC. If so, why do you synthesize ADRWC/Fe3O4? The authors' response doesn't satisfy the main purpose/objective of the study.
3) Point 7: Line 70-78. Describe the aims and objectives of study rather than methodology. The authors failed to present this point, which is the core and novelty of the current study.
4) Point 10: During carbonization, how do you provide airfree environment/anaerobic conditions? Again, this is the main methodology part, which is not addressed and if this is wrong, then all the manuscript is just a fake story.
5) Point 17: Overall, presentation of data in graphs is not appropriate. Why authors represents all graphs in the of concentration why not adsorption capacities. Revise all graphs and report results accordingly. When I gave this comment, then that means apply adsorption kinetics/isotherms/thermodynamic model in the graph and then present which again authors fail.
Author Response
Greetings dear editor,
In the box below is attached the response to the reviewer’s comments.

Round 3
Reviewer 1 Report
The manuscript has been significantly improved and can now be accepted for publication in Water.